# Treatment outcomes of diabetic ketoacidosis among diabetes patients in Ethiopia. Hospital-based study

Gizework Alemnew Mekonnen[1]*, Kassahun Alemu Gelaye[2], Eyob Alemayehu Gebreyohannes[1], Tadesse Melaku Abegaz[1]

1 School of pharmacy, College of Medicine and Health Science, University of Gondar, Gondar, Ethiopia, 2 Department of Epidemiology and Biostatistics, Institute of Public Health, College of Medicine and Health Science, University of Gondar, Gondar, Ethiopia

* gizeworkalemnew@gmail.com

## Abstract

### Background

There was limited data on treatment outcomes among patients with diabetic ketoacidosis (DKA) in Ethiopia.

### Objective

The aim of the study was to determine the treatment outcomes of DKA patients attending Debre Tabor General Hospital.

### Method

A retrospective study was conducted at Debre Tabor General Hospital and data were collected from June 1 to June 30 of 2018. Participants included in the study were all diabetic patients with DKA admitted from August 2010 to May 31, 2018. The primary outcomes were the treatment outcomes of DKA including (in-hospital glycemic control, the length of hospital stay and in-hospital mortality). The statistical analysis was carried out using Statistical Package for Social Sciences (SPSS) version 22. Descriptive statistics was presented in the form of means with standard deviation and binary regression was conducted to determine factors that affect length of hospital stay among DKA patients.

### Result

387 patients were included in the study. The mean age of patients was 33.30± 14.96 years. The most common precipitating factor of DKA was new onset diabetes mellitus 150(38.8%). The mean length of hospital stay was 4.64(±2.802) days. The mean plasma glucose at admission and discharge was 443.63(±103.33) and 172.94 (±80.60) mg/dL, respectively. The majority 370 (95.60%) of patients improved and discharged whereas 17 (4.40%) patients died in the hospital. Patients with mild and moderate DKA showed short hospital stay; AOR: 0.16 [0.03–0.78] and AOR:0.17[0.03–0.96] compared with severe DKA. Diabetic ketoacidosis precipitated by infection were nearly five times more likely to have long hospital

**Data Availability Statement:** All relevant data are within the paper and its Supporting Information files.

**Funding:** This study did not receive funding.

**Competing interests:** The authors have declared that no competing interests exist.

**Abbreviations:** DTGH, Debre Tabor General Hospital; DKA, Diabetic Ketoacidosis; DM, Diabetes Mellitus; SD, Standard Deviation; SPSS, Statistical Package for Social Sciences; USA, United States of America.

stay than DKA precipitated by other causes; AOR: 4.59 [1.08–19.42]. In addition, serum glucose fluctuation during hospitalization increased the likelihood of long hospital stay, AOR: 2.15[1.76–2.63].

## Conclusions

New onset type 1 diabetes was the major precipitating factor for DKA. Admitted DKA patients remained in hospital for a duration of approximately five days. About five out of hundred DKA patients ended up with death in the hospital. Infection, serum glucose fluctuations and severity of DKA were determinants of long hospital stay. Early prevention of precipitating factors and adequate management of DAK are warranted to reduce length of hospital stay and mortality.

## Introduction

Diabetes mellitus (DM) represents a group of metabolic disorders characterized by increased blood glucose concentration. The international diabetes federation estimated that 463 million adults were diagnosed for DM in 2019 [1–4]. Diabetic ketoacidosis (DKA) is an acute life-threatening complication of DM. Multiple pathophysiologic factors have been postulated for the pathophysiology of DKA including oxidative stress and pro-inflammatory cytokines (i.e., tumor necrosis factor-alpha (TNF-α)) that might lead to inadequate insulin secretion or utilization in the body [5–10]. The clinical characteristics of DKA include polyuria, polydipsia, weight loss, vomiting, dehydration, fatigue, mental status change, Kussmaul respirations, tachycardia, and hypotension [11, 12]. A diagnosis of DKA is made when patients are presented with blood pH level of less than 7.30, and bicarbonate level below 18 meq/L along with certain level of mental status impairment [11, 13–15].

Diabetic ketoacidosis is associated with high mortality rates in the developing world [16, 17]. The poor management of DKA can lead to debilitating and potentially fatal complication including cerebral edema and severe hypoglycemia. Mortality of DKA has been reported to be less than 5% in treatment experienced centers of the Americas, Europe and Asia [18, 19]. In Africa, the mortality of DKA is unacceptably high with a reported death rate of 26 to 29% in studies from Kenya, Tanzania, and Ghana [8]. In Ethiopia, mortality from DKA was found be high [20]. A retrospective study conducted at Shashemene Referral Hospital reported that DKA contributed 12% in-hospital mortality [21]. Another study conducted in Hiwot Fana Specialized University Hospital indicated that about 11% of patients with diagnosis of DKA died in hospital [22].

In order to reduce mortality different countries have been undertaking different strategies and prevention measures including diabetes self-management education, increasing the pathophysiology of DKA and adoption of DKA treatment guidelines [19]. However, these strategies have not been appropriately implemented in Ethiopia. In addition, the cost and lack of medication supplies, presence of comorbid conditions, inappropriate insulin storage, medication non-adherence, electrolyte disturbance and smoking habits complicated the prevention and treatment of DKA in Ethiopia [21, 23, 24]. There was limited data on treatment outcomes including hospital stay, glycemic control during hospitalization and in-hospital mortality related to DKA in north Ethiopia. The present study will determine the treatment outcomes of DKA among patients attending general hospital in North Ethiopia. The study findings will be

relevant to improve the management of DKA, to minimise in-hospital mortality and to reduce length of hospital stay among DKA patients.

## Methods

### Study setting and period

This study was conducted at Debre Tabor General Hospital (DTGH) from 1 June 2018 to 30 June 2018. Debre Tabor General Hospital is a government hospital located in Debre Tabor Town, North Ethiopia. The hospital provides twice weekly outpatient services for chronic illness patients including DM, Asthma and Cardiovascular disorders. Outpatient follow-up of DM patients and in-patient management of DKA have been undertaken in the hospital. Insulin administration and fluid resuscitation are the major emergency management of DKA along with treatment of the precipitating factors. The emergency unit of the general hospital delivers the acute management of DKA followed by in-patient service until the patient status indicate improvement in terms of functional and clinical parameters.

### Study design and population

A hospital based retrospective study was conducted on DKA patients presented to DTGH. All DKA patients who were admitted to inpatient ward from August 1, 2010, to May 31, 2018, were included in the study.

### Inclusion and exclusion criteria

**Inclusion criteria.** Participants included in the study were all adult (age≥18 years old) diabetic patients with DKA.

**Exclusion criteria.** Pregnant and breastfeeding women were excluded from the study.

### Sample size determination and sampling technique

Patients who fulfilled the inclusion criteria were selected using Convenience sampling technique.

### Study variables

**Dependent variables.** Length of hospital stay, glycemic control and in-hospital mortality.

**Independent variables.** Age, gender, residence, family history of DM, types of DM, severity of DKA, blood glucose level, blood pressure, respiratory rate, pulse rate, co-morbidities, and precipitating factors.

### Data collection methods

Medical record of patients with DKA admitted to the hospital was traced from patient logbook and drawn from card room. The data were collected by trained data collectors using structured pretested data extraction tool. Data was collected on demographics, presenting symptoms, precipitating causes of DKA, vital signs, biochemical profiles (admission blood glucose, admission urine ketone, urine glucose), length of hospitalization and in-hospital mortality.

### Data quality control technique

Data collectors were trained intensively on the contents of the questionnaire, data collection methods and ethical concerns. In order to ensure its quality, the questionnaire was pretested in 5% of DM subjects who were attending DTGH. The findings of the pretest were not

included in the final analysis. The necessary amendment was done on the final version of the questionnaire during the process of the pretest. Expert consultation was sought after developing the questionnaire for validation. Supervision and checking was made by the supervisor to ensure the completeness and consistency of the collected data. All collected data were examined for completeness and consistency during data management, storage, and analysis

## Data analysis

All the statistical data was carried out using Statistical Package for Social Sciences (SPSS), version 22 (SPSS Inc., Cary, NC, USA) [25]. Descriptive statistics was calculated for categorical variables. Bivariate analysis was done and variables with a P–value <0.2 were considered statically significant and analyzed using multivariate logistic regression. Binary logistic regression was done to determine factors that affect length of hospital stay. Variables with p–value of less than 0.05 with 95% confidence interval were considered statistically significant. Hosmer–Lemeshow goodness- of -fit test was done to check model fitness.

## Operational definitions

- **Hyperglycemia** is defined as random plasma glucose >200 mg/dL and

- **Hypoglycemia** is defined as a blood glucose level ≤70 mg/Dl [26, 27].

- **Euglycemia** is defined as serum glucose level between 100 and 200mg/dL [11].

- **Long-hospital stay** was defined as hospital stay for more than seven days [28].

- **Short-hospital stay** was defined as the patient stayed in the hospital for ≤ 7 days [29].

- **Treatment outcomes** defined as a composite of level of glycemic control, the length of hospital stay and in-hospital mortality of DKA.

- **Good glycemic control:** Proportion of patients having plasma glucose <250 mg/dL within 6 hours of therapy

- **Poor glycemic control:** Proportion of patients having plasma glucose >250 mg/dL after 6 hours of therapy.

- **Mild DKA** is defined as (arterial pH, 7.25–7.30 and serum bicarbonate, 15–18 mEq/L).

- **Moderate DKA** is defined as (arterial pH, 7.00 to < 7.25 and serum bicarbonate, 10 to < 15 mEq/L).

- **Severe DKA** is defined as (arterial pH, < 7.00and serum bicarbonate, < 10 mEq/L) [11, 18].

- **Rebound Ketonuria:** It is defined as the increase in ketone level once it is controlled with insulin management and fluid resuscitation

## Results

### Sociodemographic characteristics of DKA patients

A total of 387 patients were included in the study. Out of them, 305 (78.8%) and 82(21.2%) patients had T1DM and T2DM, respectively. The mean age of the patients was 33.30± 14.96 (range 15–64 years). More than half of the patients were urban residents. Family history of diabetes was reported in 50 (12.9%) of patients. The mean duration of DM was 26.21 (±39.62) months (Table 1).

**Table 1. Sociodemographic and disease status of DKA patients admitted at DTGH from August 2010 to May 31, 2018(n = 387).**

| Characteristics | Frequency (%) |
|---|---|
| Age | |
| Overall (Mean ±SD) | 33.29 (±14.96) |
| 15–24 | 135(34.9) |
| 25–34 | 95(24.5) |
| 35–44 | 76(19.6) |
| 45–54 | 44(11.4) |
| 55–64 | 37(9.6) |
| Sex | |
| Male | 143(37.00) |
| Female | 244(63.00) |
| Residence | |
| Urban | 264(68.20) |
| Rural | 123(31.80) |
| Family history of diabetes | |
| Yes | 50(12.90) |
| No | 53(13.70) |
| Unknown | 284(73.40) |
| Types of DM | |
| Newly diagnosed T1DM | 146(37.70) |
| Newly diagnosed T2DM | 21(5.40) |
| Known T1DM | 159(41.10) |
| Known T2DM | 61(15.80) |
| Duration of DM (mean± SD) | 26.22(±39.67) |

## Clinical characteristics of DKA patients

Polyuria and polydipsia (97.9%) were the most frequent clinical manifestations of DKA patients followed by easy fatigability (82.9%) and abdominal pain (47.0%). The mean pulse rate, systolic blood pressure (SBP), diastolic blood pressure (DBP), respiratory rate (RR) and body temperature at admission were 94.17(±14.54),104.30(±15.51), 67.84(±10.42), 24.85 (±4.30) and 36.47(±0.94), respectively. The mean frequency of DKA episodes was 1.5 times (maximum frequency of DKA episode was 8 times and the minimum was only once) since the first diagnosis of DM (Table 2).

## Precipitating factors of DKA

About 258(66.67%) of DKA patients had known precipitating factor for DKA. The predominant precipitating factor of DKA was new onset T1DM 150(38.8%) followed by poor compliance to antidiabetic treatment (14.7%) and infections (13.2%), respectively. About 129 (33.33%) patients had no known precipitating factor (Fig 1). Urinary tract infection was the most common infection 24(47.05%) that precipitated DKA and respiratory tract infection 21 (41.17%) was the second most common infection that precipitate DKA.

## Severity of DKA and ketone bodies

About three-fourth (75.5%) of the patients were presented with mild DKA, 74(19.1%) were presented with moderate DKA and the remaining 21(5.4%) were diagnosed with severe DKA.

**Table 2. Clinical characteristics of DKA patients admitted at DTGH from August 2010 to May 31, 2018(n = 387).**

| Clinical characteristics | Frequency (%) |
|---|---|
| Frequency of DKA episode | |
| once | 272(70.30) |
| twice | 74(19.10) |
| Three times | 20(5.20) |
| ≥ four times | 21(5.50) |
| Polyuria and Polydipsia | 379 (97.90) |
| Fatigability | 321(82.90) |
| Abdominal pain | 182(47) |
| Vomiting | 70(18.10) |
| Others* | 36(9.3) |
| Pulse rate (mean±SD) | 94.17(±14.54) |
| Normal | 276(71.30%) |
| Tachycardia | 108(27.90%) |
| Bradycardia | 3(0.8%) |
| Temperature ($^0$C) | 36.47(±0.94) |
| Respiratory rate (mean± SD) | 24.85(±4.300) |
| Respiratory status | Frequency (%) |
| Normal | 261(67.40) |
| Tachypnea | 126(32.60) |
| SBP (mean± SD) | 104.30(±15.510) |
| DBP (mean± SD) | 67.84(±10.422) |
| Hypotension | 46(11.9) |
| Normal | 227(58.7) |
| Elevated | 14(3.6) |
| Stage 1 | 73(18.9) |
| Stage 2 | 27(7.0) |
| Urine Ketone | |
| <+3 | 224(57.88) |
| ≥+3 | 163(42.11) |
| Urine glucose | |
| Free | 2(0.50) |
| +1 | 22(5.70) |
| +2 | 161(41.60) |
| +3 | 130(33.60) |
| +4 | 33(8.50) |
| Not measured | 39(10.10) |
| Comorbidities | |
| No | 355(91.70) |
| Yes | 32(8.30) |

*Kussmaul respiration, headache, shortness of breath, delirium, cold extremities, coughs, blurring of vision, vertigo, tinnitus, and sweating.

Around (17.6%) patients experienced rebound ketonuria once and (1.6%) patients had rebound ketone two times after hospital admission. The average duration of time required for complete removal of ketonuria was 9.23(±10.954) hours (Table 3).

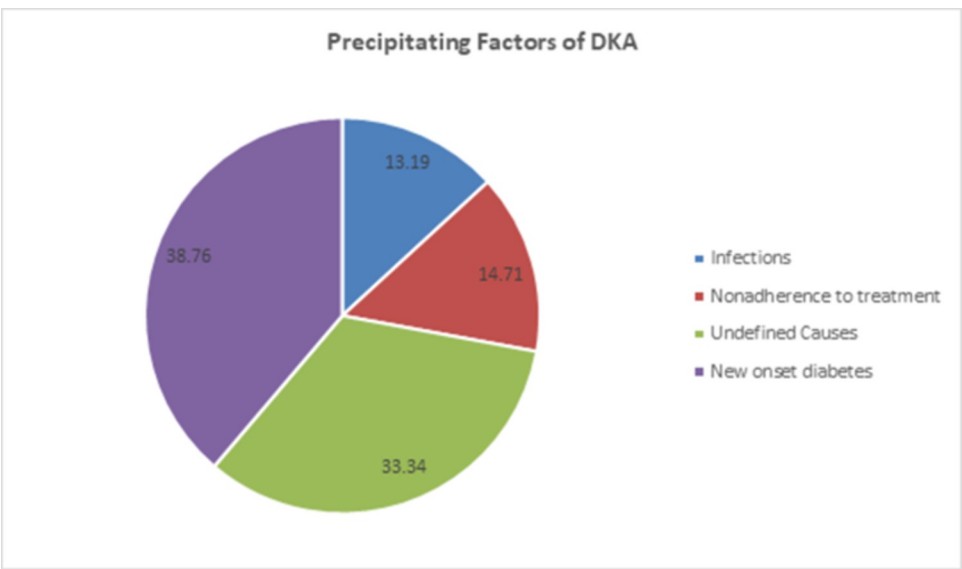

**Fig 1. Precipitating factors of DKA patients admitted at DTGH from August 2010 to May 31, 2018(n = 387).**

## The level of glycemic control of DKA patients

The mean plasma glucose at admission and discharge was 443.63(±103.33) and 172.94 (±80.60) mg/dL, respectively. The frequency of serum glucose changes during hospitalization was 6.78 (±4.43) times ranging from 0 to 32 times. The average time required to attain euglycemic state was 1.89(±1.36) (range 1–11) days after admission. Nearly three-fourth of patients achieved euglycemic state within five days of admission. About 60% of patients discharged with the serum glucose within the normal range and 127(32.8%) patients were discharged with serum glucose above the normal level (hyperglycemia). Hypoglycemic episode was observed in 70(18.10%) patients. Overall, 201(51.94%) patients did not attain glucose level below 250mg/dL in six hours (Table 4).

## Length of hospital stay and mortality

The mean length of hospital stay was 4.64±2.802 days, ranging from (1–18) days. About 79 (20%) patients had long hospital stay (>7days). Majority 370 (95.60%) of patients improved and discharged whereas 17 (4.40%) patients died in the hospital.

**Table 3. Severity of DKA and ketone bodies among DKA patients admitted at DTGH from August 2010 to May 31, 2018(n = 387).**

| Severity of DKA and urine ketone bodies | Frequency (%) |
|---|---|
| Mild DKA | 292(75.50) |
| Moderate DKA | 74(19.10) |
| Severe DKA | 21(5.40) |
| Frequency of ketone fluctuation | |
| No ketone body fluctuation | 310(80.10) |
| Once | 68(17.60) |
| Two times | 6(1.60) |
| Three times | 1(0.30) |
| Four times | 1(0.30) |
| Five times | 1(0.30) |
| Time required for Ketone free state (hours) | 9.23(±10.95) |

**Table 4. Glucose level during hospitalization in DKA patients at DTGH from August 2010 to May 31, 2018 (n = 387).**

| Glucose level during hospitalization | Frequency (%) |
|---|---|
| Glucose level at admission | 443.63(±103.33) |
| Glucose level at discharge | 172.94(±80.60) |
| Frequency of glucose level change | 6.78(±4.43) |
| Time require to achieve euglycemia (days) | 4.37(±2.68) |
| Maximum glucose values | 263.86(±123.20) |
| Minimum glucose values | 39.67(±57.31) |
| Time for glucose to be <250mg/dL(days) | 11.08(±15.121) |
| Good glycemic control | 186(48.06%) |
| Poor glycemic control | 201(51.94%) |
| Glycemic control level at discharge | |
| Hyperglycemia | 127(32.80) |
| Euglycemic | 239(61.80) |
| Hypoglycemia | 21(5.40) |
| Frequency of hypoglycemic episodes | |
| 0 | 317(81.90) |
| 1 | 47(12.10) |
| 2 | 11(2.80) |
| 3 | 10(2.60) |
| 5 | 1(0.30) |
| ≥5 | 1(0.30) |

## Factors affecting length of hospital stay

Determinants of long hospital stay on binary logistic regression analysis were age (between 35–44), severity of DKA, presence of glucose fluctuation and infections. Patients who had mild DKA showed 84% of odds of reduction in hospital stay, AOR: 0.16 [95%CI: 0.03–0.78] and moderate DKA patients have 83% less likely to have long hospital stay; AOR: 0.17[95%CI: 0.03–0.96] than severe DKA. Patients who were within age range of 35–44 had 87.5% likely hood of reduction on long hospital stay than patients of age between 55–64 years; AOR: 0.125 [0.017–0.92]. Patients whose DKA was precipitated by infection were 5 times more likely to have long hospital stay than patients with DKA precipitated by unknown causes; AOR: 4.59 [1.08–19.42]. In addition, for every episode of the glucose level fluctuation, the likelihood of long hospital stay increased more than 2 times; AOR: 2.15[1.76–2.63] (Table 5).

## Rate of decline in hyperglycemia among DKA patients

After hospital admission, the level of glucose falls rapidly in T1DM patients while gradual decline of hyperglycemia was observed in T2DM patients. The length of hospital stay (in days) was plotted against the type of DM in Kaplan Meir analysis which indicated that the hyperglycemic episode subsided sooner among T1DM group when patients were followed for a duration of hospital stay [Log rank = 3.15, p-value = 0.03] (Fig 2).

## Discussions

Successful treatment of DKA requires a prompt correction of hyperglycemia, dehydration, ketones, and acidosis [11–13]. The management of DKA involves the administration of regular insulin via continuous intravenous infusion along with adequate fluid resuscitation and

**Table 5. Determinants of hospital stay in DKA patients at DTGH from August 2010 to May 31, 2018(n = 387).**

| Variable | Length of hospital stay | | Crude Odds Ratio | Adjusted Odds Ratio |
|---|---|---|---|---|
| | Short ($\leq$ 7 days (n = 308) | Long ($>$7 days(n = 79) | | |
| **Age** | | | | |
| **18–24** | 98 (25.32%) | 37 (9.56%) | 1.61(0.65–4.00) | 0.251 (0.03–1.80) |
| **25–34** | 79 (20.41%) | 16 (4.13%) | 0.86(0.32–2.31) | 0.40 (0.05–3.10) |
| **35–44** | 62 (16.02%) | 14 (3.16%) | 0.96(0.32–2.64) | 0.125 (0.017–0.92) * |
| **45–54** | 39 (10.07%) | 5 (1.29%) | 0.54(0.15–1.90) | 0.112 (0.01–1.00) |
| **55–64** | 30 (7.75%) | 7 (1.80%) | 1 | 1 |
| **Sex** | | | | |
| **Male** | 117 (30.23%) | 26 (6.71%) | 1 | 1 |
| **Female** | 191 (49.35%) | 53 (13.69%) | 0.80 (0.47–1.35) | 2.162 (0.87–5.31) |
| **Residence** | | | | |
| **Urban** | 219 (56.58%) | 45 (11.62%) | 1.85(1.12–3.09) * | 0.446 (0.18–1.093) |
| **Rural** | 89 (22.99%) | 34 (8.78%) | 1 | 1 |
| **Type of DM** | | | | |
| **Type 1** | 237 (61.24%) | 68 (17.57%) | 0.54(0.27–1.07) | 1.028 (0.19–5.30) |
| **Type 2** | 71 (18.34%) | 11 (2.84%) | 1 | 1 |
| **History of DM** | | | | |
| **New** | 123 (31.78%) | 44 (11.36%) | 0.52(0.32–0.87) * | 0.76 (0.06–8.68) |
| **Known DM** | 185 (47.80%) | 35 (9.04%) | 1 | 1 |
| **Frequency of DKA** | 1.53 ± 1.00 | 1.37 ± 0.98 | 0.82(0.60–1.11) | 1.02(0.62–1.68) |
| **Pulse rate** | 93.94 ± 14.48 | 95.03 ±14.84 | 1.00(0.98–1.02) | 1.00 (0.97–1.04) |
| **SBP** | 104.92 ± 15.76 | 101.90 ±14.32 | 0.98(0.96–1.00) | 0.99(0.95–1.03) |
| **DBP** | 68.36 ± 10.30 | 65.82 ± 10.69 | 0.97(0.95–1.00) | 0.98(0.93–1.03) |
| **Respiratory rate** | 24.91 ± 4.180 | 24.65 ± 4.764 | 0.98(0.92–1.04) | 0.99(0.89–1.10) |
| **Temperature** | 36.47 ± 0.94 | 36.44 ± 0.93 | 0.96(0.74–1.25) | 1.08(0.70–1.66) |
| **Duration of DM** | 27.46 ± 40.39 | 21.33 ± 36.33 | 0.99 (0.98–1.00) | 0 .99 (0.98–1.01) |
| **Severity of DKA** | | | | |
| **Mild** | 237 (61.24%) | 55 (14.21%) | 0.31(0.12–0.77) * | 0.16(0.03–0.78)* |
| **Moderate** | 59 (15.24%) | 15 (3.87%) | 0.34(0.12–0.95) * | 0.17(0.03–0.96)* |
| **Severe** | 12 (3.10%) | 9 (2.32%) | 1 | 1 |
| **Comorbidities** | | | | |
| **Yes** | 28 (7.23%) | 4 (1.03%) | 1 | 1 |
| **No** | 280 (72.35%) | 75 (19.37%) | 0.53(0.18–1.56) | 1.77(0.34–9.14) |
| **Precipitating factors** | | | | |
| **Infection** | 39 (10.07%) | 12 (3.10%) | 1.78(0.79–4.00) | 4.59(1.08–19.42) * |
| **Omission** | 50 (12.91%) | 7 (1.80%) | 0.81(0.32–2.05) | 0.91(0.17–4.81) |
| **New DM** | 109 (28.16%) | 41 (10.59%) | 2.17(1.18–3.98) * | 3.46(0.39–30.81) |
| **Other** | 110 (28.42%) | 19 (4.90%) | 1 | 1 |
| **RBS** | 435.68 ± 103.15 | 474.61±98.717 | 1.004(1.001–1.006) * | 1.00(0.99–1.00) |
| **Admission urine ketone** | | | | |
| **Urine ketone <+3** | 182 (47.02%) | 42 (10.85%) | 1.27(0.77–2.09) | 1.86(0.74–4.66) |
| **Urine ketone ≥+3** | 126 (32.55%) | 37 (9.56%) | 1 | 1 |
| **Number of times serum glucose fluctuates** | 5.33 ± 2.84 | 12.46±4.93 | 1.85(1.61–2.15) ** | 2.15(1.76–2.63) ** |
| **Serum glucose at discharge** | 171.99 ± 80.40 | 176.62±81.78 | 0.648(0.99–1.004) | 1.002 (0.99–1.007) |
| **The frequency of ketone fluctuation** | 0.22 ± 0.50 | 0.32±0.72 | 1.32(0.88–1.96) | 0.638 (0.03–1.36) |

* p<0.05

** p <0.01, RBS = Random blood sugar

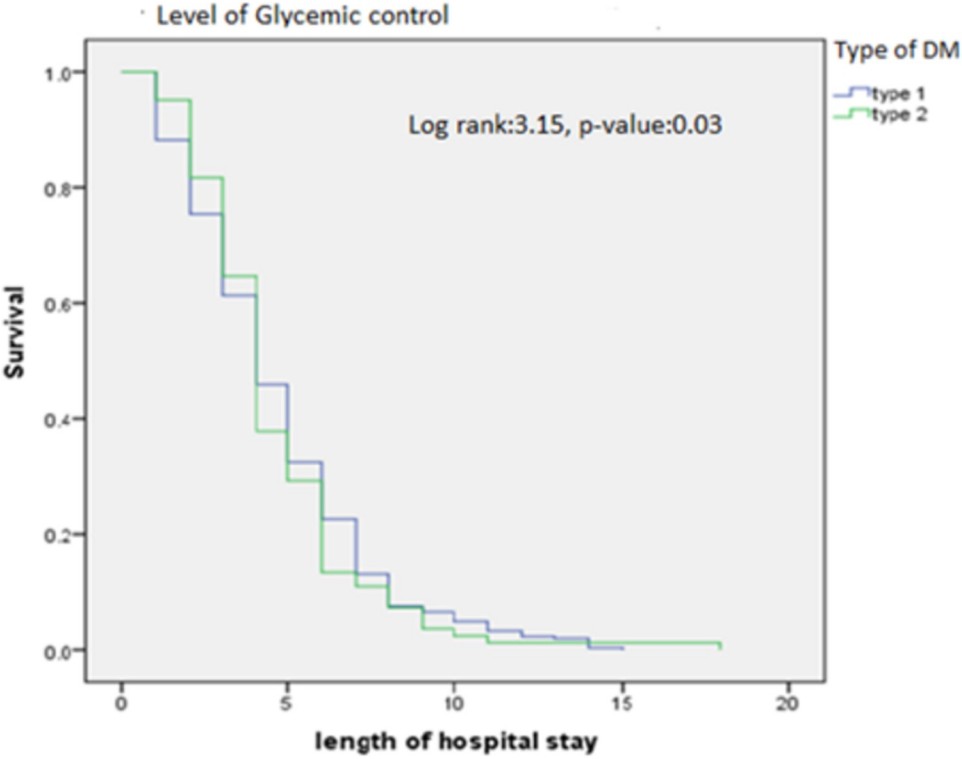

**Fig 2. Rate of decline in hyperglycemia among DKA patients.**

treatment of precipitating factors. This ultimately helps to lower the blood glucose below 250 mg/dL within 6 hours of initiation of treatment [11, 30, 31]. But the findings of this study showed that the average time required to lower plasma glucose below 250mg/dl was 11 hours which is longer than the recommended time of achieving normoglycemia. The reason for this long duration of time for the attainment of goal glucose level could be a frequent serum glucose fluctuation after hospitalization. In our study, patients had a serum glucose fluctuation of more than seven times. Patients' serum glucose increased by a maximum 263 mg/dl and a minimum of 40mg/dL above the admission glucose. Correcting this wide range fluctuation of serum glucose might take long time to obtain euglycemic state.

Our study depicted that the most frequent clinical presentations were polyuria and polydipsia (97.9%). A cross-sectional study conducted at Dilla referral hospital reported that DKA patients were commonly presented with dry mouth (30.2%) followed by altered mentation (27.8%) [32]. Our finding was in line with a study done in Libya and Nigeria [33–35]. On the contrary abdominal pain appeared to be the most common presenting symptoms in a study conducted in Egypt [36]and vomiting was the major clinical presentation in a research done at Saudi Arabia [37]and India [38]. The frequent occurrence of these symptoms in DKA patients could be due to the presence of insulin deficiency and hyperglycemia that results osmotic diuresis which in turn resulted in polyuria and polydipsia [39].

Our study showed that new onset type 1 DM was the most common precipitating factor (38.8%) followed by poor compliance to antidiabetic treatment (14.7%). New onset type 1 DM was the major precipitating factor of DKA in the present study due to the fact that our sample population constituted large number of type 1 diabetes patients in which majority of them presented with first incidence of the disease with cardinal DKA due to absolute insulin deficiency

unlike T2DM patients [40]. Another study in southwestern Ethiopia reported that the most common precipitants of DKA were infections (59%), non-compliance to medications (32.3%), and newly diagnosed diabetes (23.6%), respectively [41]. A study conducted in children's hospital in Addis Ababa revealed that the most common precipitating factors were infection (52%), omission of insulin (16%) and parasitic infection (12%) [42]. Our findings are in line with a study done in sub-Saharan Africa in which the major precipitants of DKA were new onset diabetes, missed insulin doses and infections [3]. On the other hand missed insulin dose were the most common precipitating factor in Zambia [16], Saudi Arabia (2015) [37], Libya [33]. Nairobi [43], New Zealand [44], Brazil [45], and Israel [46].

Infection was also found to be the frequent reason for the incidence of DKA in different studies. A research done in Jimma, Egypt, India, Pakistan, South Africa and Bangladesh revealed that infection was the major precipitant of DKA [20, 36, 38, 47–50]. Among infections urinary tract infection 24(47.05%) was the most common infection that precipitated DKA in our study. On the contrary respiratory tract infections were predominant infections that precipitated DKA in Libya (2007) [33], and Kenya [43]. As of the reason for the occurrence of frequent urinary tract infections, it can be due to the fact that females patient took the large proportion of our sample population in whom urinary tract infection remain the most common infection [51, 52]. Residing in developing country where hygiene is a significant problem could also favor along with comfortable condition for UTI in DM patients. Attention should be sought on self-management and personal hygiene to ultimately prevent infection.

Also, our study showed that drug non-adherence had significant contribution for DKA. Non-adherence may be due to scarcity of the antidiabetic medications or lack of awareness on how to use the drugs [53]. Therefore, it is recommended that those medications with the lowest possible price should be availed and awareness should be created on diabetic self-management and the importance of adherence in preventing the occurrence of diabetic complications.

Our study revealed that the estimated mean length of hospital stay was around five days. A cross-sectional study conducted at Jima University Hospital reported a median length of hospital stay of 6 days [41]. Our finding is lower than a study done in sub Himalayan region, Pakistan, South Africa and Saudi Arabia in which the estimated hospital stay was around 7–9 days [6, 37, 48, 49]. The hospital stay was longer when compared with Australia and New Zealand [54]. This discrepancy might be the due to the better management of the precipitating factors, adequate management of hyperglycemia and complications in developed countries.

Many factors were found to influence the length of hospital stay in the present study. It was indicated that the length of hospital stay was affected by age, severity of DKA, number of times serum glucose fluctuations and infections. Patients with mild DKA had shorter length of stay than patients with severe DKA. A study done at Israel [46] and Libya [33] showed that length of hospital stay was worse in the severe DKA. Similarly, a study done in Bangladesh showed that patients with mild DKA took short duration to recover than severe DKA [55]. Patients with severe DKA are frequently presented with severe acidotic state and severe alteration in the level of consciousness [5, 11]. This might cause complications [56] and correcting severe DKA requires long period of time which might be lead to long hospital stay in severe DKA patients. Our study also showed that patients whose DKA precipitated by infection had long hospital stay than those precipitated by undefined precipitants. This result was in line with a research done India [57]. The reason might be stress hormones secretion during infection and had detrimental effects on the immune system [58]. Consequently, patients with compromised immune system might require long period to recover. In addition, infection treatment also requires long duration of hospitalization.

In the present study 4.4% of patients died in the hospital which is almost equal with a finding in Israel 4.1% [46]. The high rate of mortality in our study might be due to high prevalence of infection which was approximately 13.2% and treatment complications including hypoglycemia as well as comorbidities. The mortality of DKA is unacceptably high with a reported death rate of 26 to 29% in studies from Kenya, Tanzania, and Ghana [28]. The cost of medications and lack of medication supplies might have contributed for the high mortality rates in Africa.

In general, the present study highlighted the treatment outcomes of DKA in terms of in-hospital mortality, glucose control and length of hospital stay. However, the study did not reveal the overall management of DKA including fluid resuscitation, and insulin administration. In addition, the retrospective nature of the study design could not enable to evaluate all outcomes of DKA. Our study did not include DKA patients whose age less than 18 years, which was the most vulnerable group of individuals for type 1 DM. Another study could be conducted to evaluate the overall trend of DKA management and outcomes in prospective approach. Part of this manuscript was presented as a preprint to research square and bioRxiv but not published in any journal as original article [59, 60].

## Conclusion

In this study, new onset type 1 DM was the major precipitating factor for DKA flowed by non-adherence. Polyuria and polydipsia were the most common presenting clinical characteristic of DKA. The mean length of hospital stay and in-hospital mortality were around 5 days. Glycemic control was poor among patients admitted with DKA. Age, infection, frequent plasma glucose fluctuation and severity of DKA were the major determinants of long hospital stay.

It is recommended that non-adherence to medications should be minimized through patient education to reduce the burden of DKA. In addition, infection preventions would attenuate the frequent incidence of DKA and its untoward impact on the lives of DM patients.

Inconsistent serum glucose level demonstrates the urgent administration of appropriate dose of regular insulin as soon as patients are diagnosed for DKA. This enables to shorten the duration of hospital stay. In the meantime, serum glucose should be monitored every hour and hypoglycemic episodes should be prevented by administrating dextrose.

## Supporting information

**S1 Data.**
(SAV)

## Acknowledgments

The authors would like to acknowledge University of Gondar and Debre Tabor General Hospital for the overall support. Part of this manuscript was presented as a preprint to research square and bioRxiv but not published in any journal.

**Declaration**

**Ethical considerations**. The study was approved by the institutional review board of University of Gondar, College of Medicine and Health Sciences with reference number 113/UoG/2018. Permission was obtained from Debre Tabor General Hospital. No patient consent was required as data was collected retrospectively. Data was collected anonymously.

## Author Contributions

**Conceptualization:** Gizework Alemnew Mekonnen, Tadesse Melaku Abegaz.

**Data curation:** Gizework Alemnew Mekonnen.

**Formal analysis:** Gizework Alemnew Mekonnen, Eyob Alemayehu Gebreyohannes, Tadesse Melaku Abegaz.

**Investigation:** Gizework Alemnew Mekonnen.

**Methodology:** Kassahun Alemu Gelaye, Eyob Alemayehu Gebreyohannes, Tadesse Melaku Abegaz.

**Supervision:** Kassahun Alemu Gelaye, Eyob Alemayehu Gebreyohannes, Tadesse Melaku Abegaz.

**Writing – original draft:** Gizework Alemnew Mekonnen.

**Writing – review & editing:** Kassahun Alemu Gelaye, Eyob Alemayehu Gebreyohannes, Tadesse Melaku Abegaz.

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
