## [Decision Letter · Decision Letter 0]

1 Sep 2021

PONE-D-21-20358

Treatment Outcome of Diabetic Ketoacidosis Among Diabetes Patients in Ethiopia: Hospital-based Study

PLOS ONE

Dear Dr. Abegaz,

Thank you for submitting your manuscript to PLOS ONE. After careful consideration, we feel that it has merit but does not fully meet PLOS ONE’s publication criteria as it currently stands. Therefore, we invite you to submit a revised version of the manuscript that addresses the points raised during the review process.

I have received the reports from our advisors on your manuscript which you submitted to PLOS ONE.

Based on the comments received, I feel that your manuscript could be reconsidered for publication should you be prepared to incorporate major revisions.

When preparing your revised manuscript, you are asked to carefully consider the reviewer comments below and submit a list of responses to the comments.

Editor Comments: The literature cited in this manuscript has not been appropriately cited. I suggested the authors to revise the introduction section and consider the following references (**J Pak Med Assoc**. 2021; 71(1): 286-96. https://doi.org/10.47391/JPMA.434; ***J Cell Biochem***. 2018;119(1):105-10. https://doi.org/10.1002/jcb.26174; ***J Cell Biochem***. 2017;118(11):3577-85. https://doi.org/10.1002/jcb.26097).

 The paper should be checked by a professional speaker of English before complete acceptance.

We look forward to receiving your revised manuscript.

Kind regards,

Muhammad Sajid Hamid Akash

Academic Editor

PLOS ONE

Journal Requirements:

Reviewers' comments:

Reviewer's Responses to Questions

**Comments to the Author**

1. Is the manuscript technically sound, and do the data support the conclusions?

Reviewer #1: Yes

Reviewer #2: Yes

2. Has the statistical analysis been performed appropriately and rigorously? 

Reviewer #1: Yes

Reviewer #2: Yes

3. Have the authors made all data underlying the findings in their manuscript fully available?

Reviewer #1: Yes

Reviewer #2: Yes

4. Is the manuscript presented in an intelligible fashion and written in standard English?

Reviewer #1: Yes

Reviewer #2: No

5. Review Comments to the Author

Reviewer #1: This paper is well written and with an excellent rationale of study. The following comments must be addressed

Introduction

There was high mortality report due to DKA at international and national level. You should have included the intervention done previously at international and national level to decrease mortality due to DKA.

Methodology

For structured questionnaire, how the reliability was assessed? How the validity was ensured?

Result

. In table 1, correct as follow: type 1dm=T1DM and type 2 dm=T2DM

Discussion

Add some latest studies in the discussion section

Reviewer #2: Reviewer comments

Manuscript ID number: PONE-D-21-20358

Title of paper: Treatment Outcome of Diabetic Ketoacidosis Among Diabetes Patients in Ethiopia: Hospital-based Study

The manuscript title seems to be an interesting article in the field of diabetes and it might attract the reader’s attention. As such, minor revisions are needed before publication on PlosOne. Please find below my general and detailed comments.

General Comments:

First of all, authors should consider the services of a professional English editor to check all the manuscript (in terms of spelling, grammar, punctuation, capitalization, brackets and others right from the title up to the conclusion part)

Please check all the references! I found some that were not uniform. (Eg: No year, Some journal names are italic and some are normal, some are written in abbreviated way, some are not, even the font size, so on….)

Some words/phrases under abbreviations were not present in the manuscript ( eg: MI, CVA, HEs, HHS, UTI) and even SPSS is not well abbreviated ( SPSS stands for Statistical Manual for Social Sciences … “manual” is it appropriate? The other thing, I think also DGH ( to me better to say “ DTGH”) throughout the manuscript including tables and figures’ legends.

Abstract:

It is well written and clear without some minor revision as indicated below.

Method:

- Line “ The statistical analysis was carried out using Statistical Package for Social Sciences (SPSS).” What is the version? Please indicate it.

Result:

- DKA…. Should be written as “Diabetic ketoacidosis” when it comes after full stop.

Conclusion:

- It is oke, but what the authors recommend? I think the authors say something here what kind/s of measure/s should be taken?

Introduction:

The introduction part is clear and short except some comments below:

- Line: “The international diabetes federation estimated that 415 million adults were diagnosed for DM in 2015(2, 3).” This information is not novel enough as it is form 2015. Update with more recent and available data. Please see the IDF DIABETES ATLAS - 9th edition (2019). Numbers are different on that report.

- DKA…. When it comes first ( the same comment to the abstract under result section)

- Line “ In Ethiopia, mortality from DKA was found be high(16).” Would you present the figure as of other countries you already mentioned?

- I understand the problem. That is well but what was the importance/significance of the study? If the authors add some justifications here please.

Data analysis section:

- Line “Variables were statically significant for p –value of less than 0.05 with 95% confidence interval.” It seems result. It needs rephrasing.

Ethical considerations:

- Reference number for ethical clearance is mentioned under declaration section (what about here? Better to add here also)

- See also capitalization here ( eg: school of pharmacy, University of Gondar College of medicine and health science.)

Operational definitions:

- Some of the definitions are cited, what about the others ( eg: Long-hospital stay, Treatment outcomes, etc)

- Line “Long-hospital stay was defined as hospital stay for more than seven days and,”… is there any word or phrase that could be continued after the word “and”? if no, better to remove the word “and “ and close it with full stop.

Results:

In general, results are well described. Try to address the following comments.

- Table 1: it says “ Type of DM” better to say it Types of DM

- …type 1 dm and ….type 2 dm (please make corrections as .. type1 DM and…. type 2 DM or Capitalize them)

- Table 2: space uniformity issue

- Figure 1 ( in the legend part, it says type 1 dm…. please capitalize it)

- Under “Severity of DKA and Ketone bodies”, line “About three-fourth (75.5%) of the patients were presented with mild DKA, approximately nineteen percent 74(19.1%) were presented with moderate DKA and the remaining 21(5.4%) were diagnosed with severe DKA.” The words “nineteen percent” and again the figure “ (19.1%), the same or ? the authors can use the one.

- Table 3: it says ( n= 387), what about the other tables?

- Under “Length of hospital stay and mortality” section: what is the importance of Table 5, since, it is stated in statement form? To me, better to avoid it. Redundancy!

- Line “The majority 370 (95.60%) of patients improved and discharged whereas 17 (4.40%) patients died in the hospital (fig 2).” Should be stated after Table 5. Here also, what is the importance of Figure 2, since it is also stated as in a statement form?

- Figure 3 : The legend should be written next to the figure

Discussion:

The authors discussed the manuscript clearly. But there are some comments as indicated below which could be addressed by the authors accordingly.

- Paragraph 1: Line “ But the findings of this study showed that the average time required to lower plasma glucose below 250mg/dl was 11 hours which is longer than the recommended time of normoglycemia.” Are there other studies in other regions in agreement with the ones from Ethiopia or others? Make some discussion. The same is true for Line “In our study, patients had a serum glucose fluctuation of more than seven times. Patients’ serum glucose increased by a maximum 263 mg/dl and a minimum of 40mg/dL above the admission glucose.”

- Paragraph 2 (clinical presentation) is well discussed, but why the authors did not compare with other studies if there in Ethiopia?

- Again paragraph 3 (new onset type 1 DM) is well discussed, but still what about the other studies done in other parts of Ethiopia? What other findings say?

- Look paragraph 5, Line “New onset type 1 DM was the major precipitating factor of DKA in the present study due to the fact that our sample population constituted large number of type 1 patients in which majority of them presented with first incidence of the disease with cardinal DKA due to absolute insulin deficiency unlike type 2 DM patients (40).” This sentence might have a greater interest at the end of the paragraph 3, where the new onset type 1 DM of the work is discussed.

- Paragraph 6 (mean length of hospital stay) is well discussed by comparing with other parts of the world, but like above comments if there are other studies in Ethiopia? Again here, references (5, 35, 36) should better put after South Africa or before the word “and”. Because Saudi Arabia seems not cited…. But there is reference 27 for it I think.

- In the last paragraph, there are some sentences seem the limitations of the study which could be better stated under “Strength and limitation of the study” section. Please try to see those sentences in this paragraph.

6. PLOS authors have the option to publish the peer review history of their article (what does this mean?). If published, this will include your full peer review and any attached files.

Reviewer #1: **Yes: **Shambel Nigussie Amare

Reviewer #2: No

---

## [Author Response · Author response to Decision Letter 0]

13 Sep 2021

Response to reviewers 

PONE-D-21-20358

Treatment Outcome of Diabetic Ketoacidosis Among Diabetes Patients in Ethiopia: Hospital-based Study

PLOS ONE 

Editor Comments: 

The literature cited in this manuscript has not been appropriately cited. I suggested the authors to revise the introduction section and consider the following references (J Pak Med Assoc. 2021; 71(1): 286-96. https://doi.org/10.47391/JPMA.434; J Cell Biochem. 2018;119(1):105-10. https://doi.org/10.1002/jcb.26174; J Cell Biochem. 2017;118(11):3577-85. https://doi.org/10.1002/jcb.26097). 

Dear Editor, 

Thank you so much for your valuable comments. We have included the suggested references in the introduction section because we have got them relevant. 

Reviewers' comments 

Reviewer #1: 

Introduction

Q1. There was high mortality report due to DKA at international and national level. You should have included the intervention done previously at international and national level to decrease mortality due to DKA. 

Dear Reviewer,

Thank you so much for your valuable comments

Answer: 

In order to reduce mortality, different countries have been using different strategies and prevention measures, such as diabetes self-management education, increasing understanding of the pathophysiology of DKA and adoption of DKA treatment guidelines. We have included this information in the introduction section. 

Methodology

Q2.For structured questionnaire, how the reliability was assessed? How was the validity ensured?

Answer;

In order to maintain the reliability and validity of the data collection tool and the collected data, pre-test was undertaken in 5% of the population and the questions were modified based on the collected data. Consultation of experts and close supervision were undertaken to further ensure the validity and reliability of the data. 

Result

Q3. In table 1, correct as follow: type 1dm=T1DM and type 2 dm=T2DM

Answer;

The necessary corrections have been made in the manuscript. 

Discussion

Add some latest studies in the discussion section 

Answer;

We have included relevant studies in the discussion section. 

Reviewer #2: 

Dear Reviewer;

Thank you so much for your valuable comments. 

General Comments:

Q1. First of all, authors should consider the services of a professional English editor to check all the manuscript (in terms of spelling, grammar, punctuation, capitalization, brackets and others right from the title up to the conclusion part). 

Answer 

The manuscript has been undergone edition by professional personnel (TMA) who is a PhD candidate at Florida A&M University, Florida, United states. 

Please check all the references! I found some that were not uniform. (Eg: No year, Some journal names are italic and some are normal, some are written in abbreviated way, some are not, even the font size, so on….)

Answer;

We have checked all references and made the necessary corrections. 

Some words/phrases under abbreviations were not present in the manuscript ( eg: MI, CVA, HEs, HHS, UTI) and even SPSS is not well abbreviated ( SPSS stands for Statistical Manual for Social Sciences … “manual” is it appropriate? The other thing, I think also DGH (to me better to say “ DTGH”) throughout the manuscript including tables and figures’ legends.

Answer;

The abbreviations were corrected and only that appeared in the manuscript were included in the list. DTGH was used in place of DGH. 

Abstract:

It is well written and clear without some minor revision as indicated below.

Method:

- Line “ The statistical analysis was carried out using Statistical Package for Social Sciences (SPSS).” What is the version? Please indicate it.

Answers

The SPSS version was included in the abstract section 

Result:

- DKA…. Should be written as “Diabetic ketoacidosis” when it comes after full stop.

Answer

DKA was written in full text when it appeared at the beginning of the sentence. 

Conclusion:

- It is oke, but what the authors recommend? I think the authors say something here what kind/s of measure/s should be taken?

Answer:

Dear reviewer;

Thank you for your comment. The recommendations were described in the main manuscript. We have also included it in the abstract section 

Introduction:

The introduction part is clear and short except some comments below:

- Line: “The international diabetes federation estimated that 415 million adults were diagnosed for DM in 2015(2, 3).” This information is not novel enough as it is form 2015. Update with more recent and available data. Please see the IDF DIABETES ATLAS - 9th edition (2019). Numbers are different on that report.

Answer;

Dear Reviewer

Thank you for your recommendation. We included the recent data on prevalence of DM replacing the old one. 

- DKA…. When it comes first (the same comment to the abstract under result section)

Answer

Corrected accordingly 

- Line “ In Ethiopia, mortality from DKA was found be high(16).” Would you present the figure as of other countries you already mentioned?

Answer

We included the figures from Ethiopia as well. 

- I understand the problem. That is well but what was the importance/significance of the study? If the authors add some justifications here please.

Answer

We have included notes in relation to the significance of the study. The study findings will be relevant to improve the management of DKA, to minimise in-hospital mortality and to reduce length of hospital stay among DKA patients.

Data analysis section:

- Line “Variables were statically significant for p –value of less than 0.05 with 95% confidence interval.” It seems result. It needs rephrasing.

Answer;

We rephrased this statement. Variables with p–value of less than 0.05 with 95% confidence interval were considered statistically significant. 

Ethical considerations: 

- Reference number for ethical clearance is mentioned under declaration section (what about here? Better to add here also)

Answer;

Reference number was included in ethical clearance section. 

- See also capitalization here ( eg: school of pharmacy, University of Gondar College of medicine and health science.)

Answer;

Corrected 

Operational definitions:

- Some of the definitions are cited, what about the others ( eg: Long-hospital stay, Treatment outcomes, etc). 

Answer;

We have included a reference for length of hospital stay. Regarding treatment outcome, we have operationalized for this research purpose, and we did not seek additional references. 

- Line “Long-hospital stay was defined as hospital stay for more than seven days and,” is there any word or phrase that could be continued after the word “and”? if no, better to remove the word “and “and close it with full stop.

Answer;

Thank you so much. We have modified according to your recommendations. 

Results:

In general, results are well described. Try to address the following comments.

- Table 1: it says “ Type of DM” better to say it Types of DM

- …type 1 dm and ….type 2 dm (please make corrections as .. type1 DM and…. type 2 DM or Capitalize them)

Answer;

We have modified the words according to your comments. 

- Table 2: space uniformity issue 

- Figure 1 (in the legend part, it says type 1 dm…. please capitalize it)

Answer;

It is important suggestion. We have modified the figure based on your recommendation 

- Under “Severity of DKA and Ketone bodies”, line “About three-fourth (75.5%) of the patients were presented with mild DKA, approximately nineteen percent 74(19.1%) were presented with moderate DKA and the remaining 21(5.4%) were diagnosed with severe DKA.” The words “nineteen percent” and again the figure “ (19.1%), the same or ? the authors can use the one.

Answer;

We have modified according to your comments. 

- Table 3: it says ( n= 387), what about the other tables?

We have included the sample sizes in each table. 

- Under “Length of hospital stay and mortality” section: what is the importance of Table 5, since, it is stated in statement form? To me, better to avoid it. Redundancy!

- Line “The majority 370 (95.60%) of patients improved and discharged whereas 17 (4.40%) patients died in the hospital (fig 2).” Should be stated after Table 5. Here also, what is the importance of Figure 2, since it is also stated as in a statement form?

We have removed the table and the figure as they are redundant. 

- Figure 3 : The legend should be written next to the figure 

The legends were written below the figure according to your recommendation. 

Discussion:

The authors discussed the manuscript clearly. But there are some comments as indicated below which could be addressed by the authors accordingly.

- Paragraph 1: Line “ But the findings of this study showed that the average time required to lower plasma glucose below 250mg/dl was 11 hours which is longer than the recommended time of normoglycemia.” Are there other studies in other regions in agreement with the ones from Ethiopia or others? Make some discussion. The same is true for Line “In our study, patients had a serum glucose fluctuation of more than seven times. Patients’ serum glucose increased by a maximum 263 mg/dl and a minimum of 40mg/dL above the admission glucose.” 

Dear Reviewer,

Thank you so much for your recommendation in relation to comparing our findings on the average time required to lower plasma glucose and glucose fluctuations. Due to the unique nature of these findings, we couldn’t find other studies however, we have discussed with the recommended guidelines regarding the time required to lower glucose within acceptable level. 

- Paragraph 2 (clinical presentation) is well discussed, but why the authors did not compare with other studies if there in Ethiopia?

Dear reviewer,

We have included other studies in Ethiopia. 

- Again paragraph 3 (new onset type 1 DM) is well discussed, but still what about the other studies done in other parts of Ethiopia? What other findings say? 

We included additional studies from Ethiopia 

- Look paragraph 5, Line “New onset type 1 DM was the major precipitating factor of DKA in the present study due to the fact that our sample population constituted large number of type 1 patients in which majority of them presented with first incidence of the disease with cardinal DKA due to absolute insulin deficiency unlike type 2 DM patients (40).” This sentence might have a greater interest at the end of the paragraph 3, where the new onset type 1 DM of the work is discussed.

We annexed the statement under paragraph 3. 

- Paragraph 6 (mean length of hospital stay) is well discussed by comparing with other parts of the world, but like above comments if there are other studies in Ethiopia? 

We included other studies in Ethiopia 

Again here, references (5, 35, 36) should better put after South Africa or before the word “and”. Because Saudi Arabia seems not cited…. But there is reference 27 for it I think.

We have corrected this statement accordingly to your comments. 

- In the last paragraph, there are some sentences seem the limitations of the study which could be better stated under “Strength and limitation of the study” section. Please try to see those sentences in this paragraph.

We revised the strength and limitation of our study and included the limitation and strength at the end of the discussion as statement form based on the journal guideline instructions. 

---

## [Decision Letter · Decision Letter 1]

15 Feb 2022

Treatment Outcomes of Diabetic Ketoacidosis Among Diabetes Patients in Ethiopia: Hospital-Based Study

PONE-D-21-20358R1

Dear Dr. Abegaz,

We’re pleased to inform you that your manuscript has been judged scientifically suitable for publication and will be formally accepted for publication once it meets all outstanding technical requirements.

Kind regards,

Paolo Magni

Academic Editor

PLOS ONE

Reviewers' comments:

Reviewer's Responses to Questions

**Comments to the Author**

The authors have adequately addressed your comments raised in a previous round of review.

Reviewer #1: All comments have been addressed

2. Is the manuscript technically sound, and do the data support the conclusions?

Reviewer #1: Yes

3. Has the statistical analysis been performed appropriately and rigorously? 

Reviewer #1: Yes

4. Have the authors made all data underlying the findings in their manuscript fully available?

Reviewer #1: Yes

5. Is the manuscript presented in an intelligible fashion and written in standard English?

Reviewer #1: Yes

6. Review Comments to the Author

Reviewer #1: (No Response)

7. PLOS authors have the option to publish the peer review history of their article (what does this mean?). If published, this will include your full peer review and any attached files.

Reviewer #1: **Yes: **Shambel Nigussie

---

## [Editor Report · Acceptance letter]

25 Mar 2022

PONE-D-21-20358R1 

Treatment Outcomes of Diabetic Ketoacidosis Among Diabetes Patients in Ethiopia. Hospital-Based Study 

Dear Dr. Mekonnen:

I'm pleased to inform you that your manuscript has been deemed suitable for publication in PLOS ONE. Congratulations! Your manuscript is now with our production department. 

Kind regards, 

on behalf of

Prof. Paolo Magni 

Academic Editor

PLOS ONE